# Deletion of *Gadd45a* Expression in Mice Leads to Cognitive and Synaptic Impairment Associated with Alzheimer’s Disease Hallmarks

**DOI:** 10.3390/ijms25052595

**Published:** 2024-02-23

**Authors:** Christian Griñán-Ferré, Júlia Jarne-Ferrer, Aina Bellver-Sanchis, Marta Ribalta-Vilella, Emma Barroso, Jesús M. Salvador, Javier Jurado-Aguilar, Xavier Palomer, Manuel Vázquez-Carrera, Mercè Pallàs

**Affiliations:** 1Department of Pharmacology, Toxicology and Therapeutic Chemistry, University of Barcelona, Avda. Joan XXIII 27, 08028 Barcelona, Spainmarta.ribalta00@gmail.com (M.R.-V.); ebarroso@ub.edu (E.B.); xpalomer@ub.edu (X.P.); pallas@ub.edu (M.P.); 2Institute of Neurosciences of the University of Barcelona, University of Barcelona, 08035 Barcelona, Spain; 3Spanish Biomedical Research Center in Neurodegenerative Diseases (CIBERNED)–National Institute of Health Carlos III, 28029 Madrid, Spain; 4Institute of Biomedicine of the University of Barcelona (IBUB), University of Barcelona, 08028 Barcelona, Spain; 5Spanish Biomedical Research Center in Diabetes and Associated Metabolic Diseases (CIBERDEM)–National Institute of Health Carlos III, 28029 Madrid, Spain; 6Pediatric Research Institute-Hospital Sant Joan de Déu, Esplugues de Llobregat, 08950 Barcelona, Spain; 7Department of Immunology and Oncology, National Center for Biotechnology/CSIC, 28049 Madrid, Spain; jmsalvador@cnb.csic.es

**Keywords:** cognitive impairment, Alzheimer’s disease, Tau pathology, neuroinflammation, *Gadd45a*

## Abstract

*Gadd45* genes have been implicated in survival mechanisms, including apoptosis, autophagy, cell cycle arrest, and DNA repair, which are processes related to aging and life span. Here, we analyzed if the deletion of *Gadd45a* activates pathways involved in neurodegenerative disorders such as Alzheimer’s Disease (AD). This study used wild-type (WT) and *Gadd45a* knockout *(Gadd45a^−/−^)* mice to evaluate AD progression. Behavioral tests showed that *Gadd45a^−/−^* mice presented lower working and spatial memory, pointing out an apparent cognitive impairment compared with WT animals, accompanied by an increase in Tau hyperphosphorylation and the levels of kinases involved in its phosphorylation in the hippocampus. Moreover, *Gadd45a^−/−^* animals significantly increased the brain’s pro-inflammatory cytokines and modified autophagy markers. Notably, neurotrophins and the dendritic spine length of the neurons were reduced in *Gadd45a^−/−^* mice, which could contribute to the cognitive alterations observed in these animals. Overall, these findings demonstrate that the lack of the *Gadd45a* gene activates several pathways that exacerbate AD pathology, suggesting that promoting this protein’s expression or function might be a promising therapeutic strategy to slow down AD progression.

## 1. Introduction

The aging process unfolds as a gradual pathophysiological journey, precipitating a diminishing state of physical and cognitive functions throughout all organs. This progressive decline is underscored by the accrual of damage in response to diverse insults and stressors [1]. Furthermore, a decreased ability to cope with those stressors is one of the main aging hallmarks [2]. Of note, aging is a driving factor for neurodegenerative diseases such as Alzheimer’s disease (AD) [3]. AD is considered the principal cause of dementia, and with the increasing population age, its prevalence tends to rise every year [4]. In particular, AD is characterized by the appearance of extracellular amyloid-beta (Aβ) plaques [5,6] and the hyperphosphorylation of Tau protein, forming intracellular neurofibrillary tangles in the brain [7,8].

Furthermore, these neuropathological accumulations are accompanied by a neuroinflammatory response, mitochondrial dysfunction, apoptosis, and autophagy, among others [9,10]. Specifically, autophagy is a relevant intracellular self-degradative process that degrades and recycles cellular components such as dysfunctional organelles and abnormally aggregated and misfolded proteins [11]. Interestingly, Tau pathology depends on the autophagy process and this mechanism is dysregulated in AD patients, facilitating protein aggregation and disease progression [12]. Together, the alterations previously mentioned cause a progressive neurodegenerative process characterized by a synaptic deficit associated with changes in the number and shape of dendritic spines, chronic oxidative stress (OS), and, finally, neuronal death [13,14].

Remarkably, under stress and cell damage conditions, the cell system attempts to recover from the harmful stimulus, partly by inducing DNA damage-inducible GADD45 proteins [15]. GADD45 proteins are essential signal transducers associated with several cellular mechanisms, including cell-cycle control, DNA damage sensation and repair, genotoxic stress, neoplasia, apoptosis, and molecular epigenetics [16,17,18]. Notably, three genes of the *Gadd45* gene family are known: *Gadd45a*, *Gadd45b,* and *Gadd45g* [19]. Each of them is expressed in different tissues, including the heart, brain, lungs, and liver. The similar structure and charge characteristics of *Gadd45* members suggest similar, but not identical, functions for these three genes [20]. Their induction promotes growth arrest, anti-inflammatory signals, apoptosis, DNA repair, genomic stability, and modulation of immune response, among others [20]. All these processes are important determinants of survival and longevity, making GADD45 members highly relevant in aging and age-related diseases [19]. Thus, due to their pleiotropic actions, a decreased inducibility of GADD45 proteins might promote consequences to the aging process and age-related disorders such as tumorigenesis, immune disorders, insulin resistance, and reduced response to stress. Interestingly, an increase in *Gadd45* genes in postmortem brain tissue obtained from patients suffering from AD has been shown [21], whereas their overexpression increases the life span in *Drosophila melanogaster* [22]. Additionally, it is known that one of the members of the *Gadd45* family, concretely *Gadd45a*, is regulated by OS and neuroinflammation through stress-responsible transcription factors, such as activating transcription factor (ATF) [23] and nuclear factor-kappa B (NF-kB) [20]. These findings indicate that *Gadd45a* dysfunction or deletion might underlie pathophysiological conditions associated with age-related cognitive decline and AD. However, the involvement of *Gadd45a* in the AD progression remains unclear.

In this study, we aimed to evaluate the role of *Gadd45a* on behavior and cognition by using the *Gadd45a^−/−^* mouse. In addition, we studied the molecular changes regarding AD hallmarks, neuroinflammation, synaptic plasticity, and autophagy processes. Interestingly, we found that lacking the *Gadd45a* gene is critical to cognition and the neurodegenerative process by increasing neuroinflammation and altering the autophagy process and synaptic plasticity.

## 2. Results

### 2.1. Gadd45a^−/−^ Mice Show Behavioral and Cognitive Deficiencies

We performed several tasks to characterize the behavioral and cognitive profile of *Gadd45a^−/−^* mice. In the OFT, we measured parameters such as locomotor activity, time in the center, time in the border, and number of rearings and groomings (Figure 1A–E). The results showed that *Gadd45a^−/−^* mice traveled less distance than the WT group, demonstrating reduced locomotor activity (Figure 1A). Furthermore, when we analyzed the time in the center and the border, we found that WT mice significantly stayed longer in the center of the box, whereas the *Gadd45a^−/−^* group spent more time in the box’s border (Figure 1B,C). Moreover, we showed that the number of rearings was higher in the KO group, whereas the number of groomings was significantly lower in this latter group of animals (Figure 1D,E).

In the NORT, we evaluated the working memory of the mice. Our results demonstrated that the *Gadd45a^−/−^* group presented a significantly lower DI compared with the WT group in both short- and long-term memory tests (Figure 1F,G). Finally, through the OLT, we analyzed spatial memory. Likewise, we found that DI was significantly lower in *Gadd45a^−/−^* mice, pointing out a deficit in spatial memory compared with the WT animals (Figure 1H).

### 2.2. Alterations in GSK3β and p44/42MAPK (ERK1/2) Promote Tau Hyperphosphorylation in Gadd45a^−/−^ Mice

To investigate if *Gadd45a* is involved in the hyperphosphorylation of Tau protein, an AD hallmark, we studied a central kinase responsible for this phosphorylation, the glycogen synthase kinase 3β (GSK3β). Thus, our analysis revealed a reduction in GSK3β phosphorylation for the Ser^9^ site, the inactivated form of the kinase, in the *Gadd45a^−/−^* mice group compared with the WT group (Figure 2A–C). On the other hand, we also evaluated another protein that mediates Tau hyperphosphorylation, the p42/p44 mitogen-activated protein kinase (MAPK). Interestingly, our results showed a tendency to increase p-p44/42 MAPK, the active form, in the KO mice group compared with the WT mice (Figure 2D–F).

Hence, considering these results, we analyzed Tau hyperphosphorylation protein levels in both groups of mice. We found an increase in Tau phosphorylation in KO animals, concretely for the Ser^396^ and Ser^404^ phosphorylation sites, compared with the WT mice group, reaching significance in the Ser^404^ phosphorylation site and suggesting a participation of *Gadd45a* in Tau pathology (Figure 2G–J).

### 2.3. Lack of Gadd45a Increases Proinflammatory Target Genes and the NF-kB Pathway

Next, we analyzed different proinflammatory markers to demonstrate that the *Gadd45a* gene could also play a role in the regulation of neuroinflammation. Strikingly, we showed a clear tendency to increase p-NF-kB protein levels in the *Gadd45a^−/−^* mice group compared with the WT group (Figure 3A–C). Concretely, this phosphorylation allows its translocation into the nucleus and promotes the expression of different genes involved in the inflammatory process. Then, we evaluated those pro-inflammatory target genes, such as *triggering receptors expressed on myeloid cells 2 (Trem2), arginase 1 (Arg1), inducible nitric oxide synthase* (*iNOS), tumor necrosis factor-alpha (Tnf-α), and interleukin-1β (Il-1β)*. For all the genes, we found a significantly higher expression in *Gadd45a^−/−^* animals compared with the WT mice group, suggesting a role for GADD45A in modulating the stressor molecular events (Figure 3D).

### 2.4. Lack of Gadd45a Gene Expression Promotes Alterations in Autophagy Markers

Finally, the protein levels of different autophagy markers were also evaluated to study whether the *Gadd45a^−/−^* mice present alterations in the autophagy process. Interestingly, our results showed a higher protein expression of p-ULK1 Ser^757^, whose phosphorylation reduces the autophagosome formation in the KO mice group rather than in the WT group (Figure 4A–C). In contrast, protein levels of Beclin-1 showed a significant reduction in the *Gadd45a^−/−^* group compared with the WT animals (Figure 4A–C). We also evaluated p62 protein levels, and our results demonstrated a significant increment in the *Gadd45a^−/−^* group compared with the WT animals (Figure 4A,C). Finally, we determined the gene expression of microtubule-associated protein 1A/1B-light chain 3 II (LC3-II), and we found a reduced expression in the *Gadd45a^−/−^* group compared with the WT animals (Figure 4D). Together, these results demonstrated that autophagic activity is impaired when *the Gadd45a* gene is knocked down, which might contribute to the neurodegenerative process presented in this transgenic mouse strain.

### 2.5. KO-Gadd45a Mice Show Reduced Synaptic Plasticity Genes Accompanied by Dendritic Morphological Abnormalities

To demonstrate a reduction in synaptic plasticity associated with cognitive decline and due to all these AD hallmarks presented in *Gadd45a^−/−^* mice, we evaluated some relevant neurotrophins such as *neurotrophin 3 (Nt3), brain-derived neurotrophic factor (Bdnf), nerve growth factor (Ngf)*, *tropomyosin receptor kinase A (TrkA), and tropomyosin receptor kinase B (TrkB)* gene expression (Figure 5A–E). Remarkably, we found a significant reduction in *Bdnf*, *TrkA*, and *TrkB* gene expression in the *Gadd45a^−/−^* group compared with the WT group (Figure 5B,D,E). Likewise, *Nt3* also presented a tendency to decrease in KO animals (Figure 5A). In contrast, we found a significantly increased expression in *Ngf* in the *Gadd45a^−/−^* mice group compared with the WT mice group (Figure 5C). Moreover, we also studied the protein levels of postsynaptic density protein 95 (PSD95) (Figure 5F,G). Strikingly, a reduction in its expression was observed in the *Gadd45a^−/−^* mice group compared with the WT group, although differences did not reach statistical significance (Figure 5F,G).

Considering these results, we decided to analyze the complexity of the neurons and the dendrite density located on the cortex of WT and KO brain animals by Golgi Staining. We found that the dendrite branching, explored through the number of intersections versus the distance from the soma, was significantly lower in KO animals compared with the WT mice, meaning that neurons in the *Gadd45a^−/−^* mice exhibited a more superficial and abnormal morphology (Figure 5H,I). Moreover, regarding dendrite density, we demonstrated a reduction in the dendritic spines in the *Gadd45a^−/−^* mice group compared with the WT group, suggesting the importance of GADD45A in the synaptic plasticity process (Figure 5J).

## 3. Discussion

In the present study, we demonstrated for the first time that deleting the *Gadd45a* gene promotes several cognitive and molecular brain alterations in mice, which results in abnormal features, like neuropathological AD hallmarks. So far, under physiological stress conditions, *Gadd45a* modulates several biological processes: promoting anti-inflammatory signals, DNA repair, and genomic stability [24,25,26]. All these processes are closely related to aging and several age-related diseases, such as AD [22]. Likewise, it has been reported that the increase in *Gadd45a* expression in brains obtained from AD patients leads to neuroprotection through different molecular pathways [21]. Furthermore, it was recently published that overexpression of the pro-longevity gene *D-Gadd45* in Drosophila neurons leads to a postponed manifestation of histological and ultrastructural features of age-dependent neurodegeneration [22]. Considering this, we hypothesized that deleting the *Gadd45a* gene exacerbates AD pathology. Consequently, we assessed the behavioral and molecular alterations that generate the lack of *Gadd45a* gene expression in mice.

First, in the OFT, we found that WT animals traveled a higher distance and expended more time in the center, which could indicate lower anxiety-like behavior levels in these animals compared with the *Gadd45a^−/−^* group. Moreover, it is known that a typical feature of AD is a severe cognitive decline accompanied by memory impairment [27]. Here, we reported that the WT mice group presented a better DI than the *Gadd45a^−/−^* group in the NORT. Likewise, WT animals exhibited a higher DI than the KO-*Gadd45a* animals in the OLT. Therefore, these behavioral tests showed that the *Gadd45a^−/−^* group presented worse cognitive performance in comparison with the WT mice group. To the best of our knowledge, this is the first time that the cognitive effects of the deletion of *Gadd45a* in transgenic mice have been described. Significantly, based on these findings, we can suggest the participation of the *Gadd45a* gene in cognition as well as in anxiety-related behavior. In line with our results, other studies also demonstrate the involvement of the other *Gadd45* family members, *Gadd45b* and *Gadd45g*, in regulating memory consolidation and formation, especially in aversive learning and spatial navigation [28,29]. These works reinforce the implication of *Gadd45a* in the learning and memory processes reported here.

Because of the cognitive alterations found in the *Gadd45a^−/−^* mice group, which are also associated with age-related cognitive decline and AD, it was interesting to evaluate some parameters related to neurodegeneration. In this way, we first analyzed Tau hyperphosphorylation, one of the main hallmarks of AD [30]. Concretely, Tau hyperphosphorylation is modulated by different protein kinases, such as GSK3β [31] and p44/42 MAPK [32]. On the one hand, GSK3β kinase phosphorylates Tau protein in serine and threonine residues [33], leading to axonal transport alterations and promoting the formation of neurofibrillary tangles (NFTs) [31]. However, GSK3β is inactivated when it is phosphorylated in Ser^9^, improving cell function and reducing NFT levels [34]. On the other hand, p44/42 MAPK kinase is also activated in the first stages of AD [35]. This activation requires the phosphorylation of its threonine and tyrosine residues by the dual-specificity of MAP kinase/ERK kinase (MEK1/2) [36]. In this study, we found that the *Gadd45a^−/−^* mice group showed lower p-GSK3β Ser^9^ protein levels and increased p-p44/42MAPK protein levels, demonstrating why KO animals presented more hyperphosphorylated Tau and, subsequently, the aforementioned behavior impairment. Next, we assessed the Tau pathology protein, which causes an increase in Tau phosphorylation in the *Gadd45a^−/−^* group compared with the WT group, specifically at the Ser^396^ and Ser^404^ phosphorylation sites. Remarkably, to the best of our knowledge, this is the first report in which the deletion of *Gadd45a* gene expression is directly connected with the increase in Tau hyperphosphorylation and, therefore, AD.

In addition to Tau hyperphosphorylation, it is well established that neuroinflammation also plays an important role in AD [37,38]. In fact, growing evidence postulates that it could be the leading cause of the disease [39]. In this way, previous studies showed the activation of microglia and astroglia and the release and increase in inflammatory mediators in patients who suffer from AD [40]. Moreover, innate system activation has been associated with several AD risk genes contributing to disease progression [37,41]. In this line, several pro-inflammatory pathways are triggered in AD through NF-κB activation [42]. NF-κB is an inducible transcription factor that mediates and regulates the expression of several genes involved in the inflammatory response. Notably, GADD45A functions as a downstream target of NF-κB, regulating several cell activities such as apoptosis, cell survival, and growth arrest [20]. As expected, we found a higher NF-κB protein expression in the *Gadd45a^−/−^* group than in the WT group. Then, we studied the expression of some of the genes activated by NF-κB, such as *Trem2*, *iNOS, Tnf-α, and Il-1β. Trem2* is a cell surface receptor expressed in microglia cells, iNOS is an enzyme that releases several inflammatory mediators and generates reactive oxygen species, and Tnf- α and IL-1β are cytokines required for activating the innate immune response. Therefore, all of the themes produce neuroinflammation followed by neuronal damage and AD progression [43,44,45]. Here, we showed that these genes were increased in KO animals. This suggests that the lack of *Gadd45a* gene expression exacerbates neuroinflammation and partially explains the cognitive impairment previously observed in this mice model.

*Gadd45a* also regulated autophagy processes in charge of clearing protein aggregates [46]. Therefore, we evaluated several autophagy markers, such as ULK1 and Beclin1, both critical proteins involved in initiating autophagy [47]. Importantly, ULK1 is inactivated when it is phosphorylated at the Ser^757^ phosphorylation site [48]. Likewise, Beclin-1 is phosphorylated and activated by ULK1, a crucial step for its autophagy [49]. In this way, we found higher protein levels of p-ULK1 Ser^757^ in the *Gadd45a^−/−^* mice model, suggesting an altered autophagic process. On the contrary, in Beclin1, KO animals showed lower protein levels than the WT group. These results supported the idea that the deletion of *Gadd45a* deregulates autophagy processes. Furthermore, we also analyzed p62 protein levels. Of note, p62 is a multifunctional protein regulated by autophagic degradation and with the capacity to bind several regulators such as NRF2, mTORC1, and NF-κB, associating p62 to OS and inflammation. We found increased protein levels in p62 in the *Gadd45a^−/−^* mice model compared with the WT mice, indicating its involvement in the alterations of the autophagic process presented in the KO mice model and promoting the neurodegenerative process [50]. In consonance with the results obtained, several reports showed that inhibition of autophagy upregulates the p62 and enhances DNA damage, OS, and inflammation [50,51]. Finally, we also evaluated *LC3-II* gene levels, which are essential for autophagosome biogenesis and maturation [52]. Our results showed that *Gadd45a^−/−^* mice presented lower LC3II/I gene levels compared with WT mice. These results are in line with other reports that demonstrated that *Gadd45a* inhibits autophagy by decreasing LC3-II expression and impairing the Beclin1 and phosphatidylinositol 3-kinase catalytic subunit type 3 (PIK3C3) interaction and complex formation, producing a reduction in the number of autophagosomes [53].

Neurotrophins are key regulators of neural survival, synaptic plasticity, and development, meaning they are involved in neuroprotection. In particular, TrkA binds to *Ngf*, whereas TrkB binds to *Bdnf* [54]. Several studies showed that the GADD45A protein regulates neuronal plasticity through multiple cellular processes, including regulating *Bdnf* in neurons [55,56]. Then, to further investigate the consequences of the lack of *Gadd45a* gene expression in synaptic dysfunction, we first evaluated different neurotrophins, such as *Bdnf, Nt3,* and *Ngf,* and their receptors, *TrkA* and *TrkB.* A reduction in both receptors and neurotrophin gene expression in KO animals was found, except for *Ngf*, which presented higher levels.

Moreover, we found lower protein levels of PSD95, a major marker of synaptic maturation and plasticity, in KO animals. Considering this synaptic dysfunction observed in the *Gadd45a^−/−^* mice model, we hypothesize that the neuronal and dendritic morphology could also be affected similarly in AD [57]. Concretely, loss of dendritic spines is closely linked with synaptic dysfunction, loss of memory, and cognitive impairment [58], which are processes that have already been analyzed in this study. Therefore, we assessed the complexity of the neurons and the dendrite density of both mice groups by Golgi staining. Accordingly with reports associating GADD45A protein with the control of synaptic plasticity and memory formation [25,49], a significant reduction in dendrite branching and density was found in the *Gadd45a^−/−^* mice group, indicating that lack of *Gadd45a* gene expression results in an abnormal neuronal morphology, then in functionality. In addition, *Gadd45a* stimulates neurite outgrowth in a neuroblastoma cell culture [59]. Noteworthy, the decreased levels of neurotrophins shown in KO animals are also related to abnormal synaptic plasticity. Taken together, our findings demonstrated that *Gadd45a* mediates neuronal plasticity.

In conclusion, deletion of *Gadd45a* promotes cognitive impairment and exacerbates AD hallmarks, deregulating several biological processes that are proper to the disease: memory impairment, neuroinflammation, synaptic plasticity, and autophagy alterations (Figure 6). Therefore, keeping in mind that in AD patients brain tissue displays increased GADD45A protein levels in the hippocampus and occipital lobe and that GADD45A has a crucial function in the response to DNA strand breakage induced by Aβ in neurons [18], our study provides robust evidence that GADD45A will become a promising new target to be considered for neurodegenerative diseases, as well as new therapies for AD.

## 4. Materials and Methods

### 4.1. Animals

Wild-type (WT, *n* = 16) and knockout (*Gadd45a^−/−^* KO, *n* = 16) 4-month-old mice were used to perform the behavioral and molecular analyses. Animals had free access to water and food and remained under standard temperature conditions (22 ± 2 °C) and 12 h/12 h light/dark cycles (300 lux/0 lux).

All the procedures with the animals, including the behavior tests and the brain dissection and extraction, followed the ARRIVE and the standard ethical guidelines (European Communities Council Directive 2010/63/EU and Guidelines for the Care and Use of Mammals in Neuroscience and Behavioral Research, National Research Council 2003) and were approved by the Institutional Animal Care and Generalitat de Catalunya (#10291, 28 January 2018). All efforts were made to minimize the suffering and the number of mice used.

### 4.2. Behavioral and Cognitive Tests

#### 4.2.1. Open Field Test

The Open Field Test (OFT) allows us to evaluate the anxiety behavior of the mice. The test was performed in a white wooden box (50 cm × 50 cm × 25 cm). Its floor was divided into two areas and defined as a center zone and a border zone (15 cm between the center zone and the wall). The mice were placed in the center of the box and allowed to explore it for 5 min. Behavior was scored with SMART^®^ ver. 3.0 software and the parameters analyzed included distance traveled, time in the center zone, time in the border zone, and number of rearings and groomings. Afterward, the mice were returned to their home cages, and the OFT apparatus was cleaned with 70% EtOH. The parameters measured in the OFT are in Appendix A.

#### 4.2.2. Novel Object Recognition Test

The Novel Object Recognition Test (NORT) allows the evaluation of the mice’s short- and long-term memory. A 90-degree two-arm (25 cm long, 20 cm high, 5 cm wide) black maze was used to perform the test. The light intensity was 30 lux and the maze walls were cleaned with 70% ethanol between trials to eliminate the olfactory cues. The objects to be discriminated were plastic figures (object A, 5.25 cm high, and object B, 4.75 cm tall). First, the mice were habituated to the maze for three days for 10 min. On day 4, the acquisition trial (first trial, familiarization) was performed, and the animals were allowed, for 10 min, to explore two identical objects (A + A, B + B) placed at the end of each arm. After 2 h (short-term memory) and 24 h (long-term memory), the retention trial (second trial) was performed, in which one of the two old objects was replaced by a new one, and the mice were allowed to explore the objects again for 10 min.

The time that the animal explored the new (TN) and the old object (TO) was measured, and a discrimination index (DI) was defined as (TN − TO)/(TN + TO). Sniffing or touching objects with the nose and forepaws was considered exploration. Objects A and B were counterbalanced to avoid object preference biases: half of the animals were exposed first to object A and the other half to object B.

#### 4.2.3. Object Location Test

The Object Location Test (OLT) allows us to evaluate the spatial memory of the mice. This test is based on the spontaneous tendency of mice to spend more time exploring a novel object location than a familiar object location, as well as to recognize when an object has been relocated. To perform the test, a wooden box (50 cm × 50 cm × 25 cm) was used, in which three walls were white and only one was black. The light intensity was 30 lux, and the box was cleaned with 70% ethanol between trials to eliminate the olfactory cues. On the first day, the animals were placed in the center of the box and allowed to explore and familiarize themselves with the box for 5 min. On day 2, two identical objects (A + A, 10 cm high) were put in front of the black wall at equal distances from each other and the wall (training phase). The mice were placed in the center of the box and allowed to explore for 5 min. On day 3, the animals were placed again in the center of the box, but one object was moved to a different position in front of the white wall. The trials were recorded with a camera mounted above the open field area, and the total exploration time was determined by evaluating the amount of time (seconds) spent sniffing the object in the new location (TN) and the object in the old location (TO). The DI was calculated to evaluate the cognitive performance, which is defined as (TN − TO)/(TN + TO).

### 4.3. Brain Processing

WT and KO mice were euthanized by cervical dislocation one day after OLT. The brains were immediately removed from the skull. Then, the hippocampus and cortex were extracted and frozen in dry ice. The samples were kept at −80 °C for further use.

### 4.4. Western Blotting (WB)

The hippocampus tissue samples (*n* = 6) were homogenized with lysis buffer (50 mM Tris-HCl, 150 mM NaCl, five mM EDTA, 1% Triton X-100, pH 7.4), EDTA-free Protease inhibitor cocktail (Roche, Mannheim, Germany), and Phosphatase inhibitor cocktail II (Sigma-Aldrich, St. Louis, MO, USA). Total protein was extracted, and the Bradford method was used to quantify the protein concentration of each sample.

For Western blotting (WB), aliquots of 20 μg of hippocampal protein were used. Protein samples were separated by sodium dodecyl sulfate–polyacrylamide gel electrophoresis (SDS-PAGE) (8–12%) and transferred onto polyvinylidene difluoride (PVDF) membranes (Millipore). Then, the membranes were blocked in 5% Bovine Serum Albumin (BSA) in 0.1% Tris-buffered saline with Tween 20 (TBS-T) for one hour at room temperature, followed by the overnight incubation at 4 °C with the primary antibodies listed in the Appendix A.

The next day, the membranes were washed with TBST-T and incubated with the secondary antibodies for one hour at room temperature. A chemiluminescence-based detection kit was used to view the immunoreactive proteins following the manufacturer’s protocol (ECL Kit; Millipore, Burlington, MA, USA), and digital images were acquired using an Amersham Imager 680 (BioRad, Hercules, CA, USA). Then, semiquantitative analyses were carried out using Image Lab software 6.1 (Bio-Rad). The results were expressed in arbitrary units (AUs), with the control protein levels set as 100%. The results for protein quantification were normalized to the control protein levels (glyceraldehyde-3-phosphate dehydrogenase, GAPDH).

### 4.5. RNA Extraction and Gene Expression Determination

Total RNA isolation from the samples was performed using Trizol reagent following the manufacturer’s instructions. RNA yield, purity, and quality were determined spectrophotometrically using a NanoDrop™ ND-1000 (Thermo Fisher Scientific, Waltham, MA, USA). The samples were also tested in an Agilent 2100B Bioanalyzer (Agilent Technologies, Santa Clara, CA, USA) to determine the RNA integrity number. RNAs with 260/280 ratios and RIN higher than 1.9 and 7.5, respectively, were selected. Then, Reverse Transcription Polymerase Chain Reaction (RT-PCR) was carried out as follows: 2 μg of messenger RNA (mRNA) was reverse-transcribed using the High-Capacity cDNA Reverse Transcription kit (Applied Biosystems, Foster City, CA, USA).

Real-time PCR (qPCR) was conducted on the Step One Plus Detection System (Applied Biosystems, Waltham, MA, USA), employing the SYBR Green PCR Master Mix (Applied Biosystems). Briefly, each reaction mixture was composed of 6.75 μL of cDNA (whose concentration was 2 μg), 0.75 μL of each primer (whose concentration was 100 nM), and 6.75 μL of SYBR Green PCR Master Mix (2X). The comparative cycle threshold (Ct) method (ΔΔCt) was used to analyze the data, where the housekeeping gene level (β-actin) was utilized to normalize differences in sample loading and preparation. The primers are listed in Appendix A. Each sample (*n* = 6) was analyzed in duplicate, and the results represented the n-fold difference in transcript levels among different groups.

### 4.6. Dendritic Length, Spine Density, and Golgi Staining Protocol

WT and KO mice were euthanized by cervical dislocation, and the whole brain was removed from the skull (*n* = 6). Then, the Golgi staining protocol was followed using the FD Rapid Golgi Stain kit according to the manufacturer’s instructions. Images of neurons for the dendritic branching analysis were taken at 20× magnification objective in an Olympus BX61 microscope coupled to an Olympus DP70 camera. The neurons’ neurite length and complexity were measured using NeuronJ macros and Advanced Sholl Analysis. The number of intersections (branch points) within concentric circles of 10 µm radius were calculated and compared between groups. Images of neurons for analyzing the spine density were performed at 50× oil objective magnification. All neurites analyzed were around 18 µm, at a maximum distance of 150 µm from the soma. 

### 4.7. Statistical Analysis

Data are expressed as the mean ± standard error of the mean (SEM). Statistical analysis was performed using GraphPad Prism version 9. Means were compared with a one-tailed Student’s *t*-test. Statistical significance was considered when *p* values were <0.05. Statistical outliers were carried out with Grubbs’ test and were removed from the analysis. Finally, a schematic representation of the experimental design is showed in Figure 7. 

## Figures and Tables

**Figure 1 ijms-25-02595-f001:**
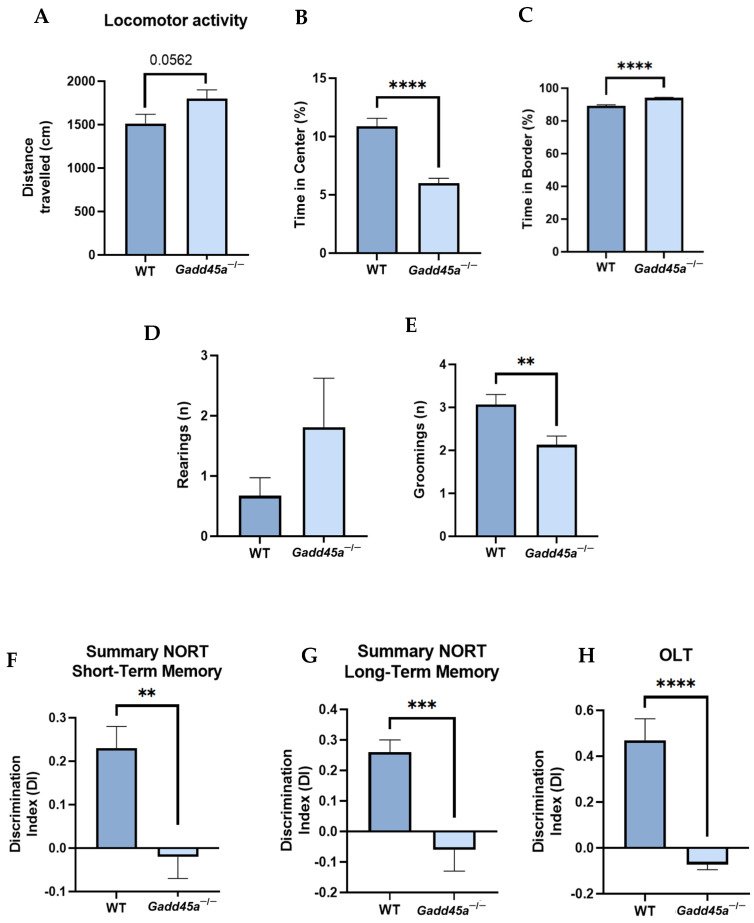
*Gadd45a^−/−^* mice behavioral and cognitive results and analysis. The results of the open field test: (**A**) distance traveled, (**B**) time spent in the central zone, (**C**) time spent in the border zone, (**D**) number of rearings, and (**E**) number of groomings. (**F**) Short-term memory test at 2 h and (**G**) long-term memory test at 24 h. (**H**) DI obtained in OLT. The results are expressed as a mean ±  SEM. One-tailed Student’s t-test compared groups; (*n* = 15 per group in WT and *n* = 15 per group in *Gadd45a^−/−^*); ** *p* < 0.01; *** *p* < 0.001; **** *p* < 0.0001.

**Figure 2 ijms-25-02595-f002:**
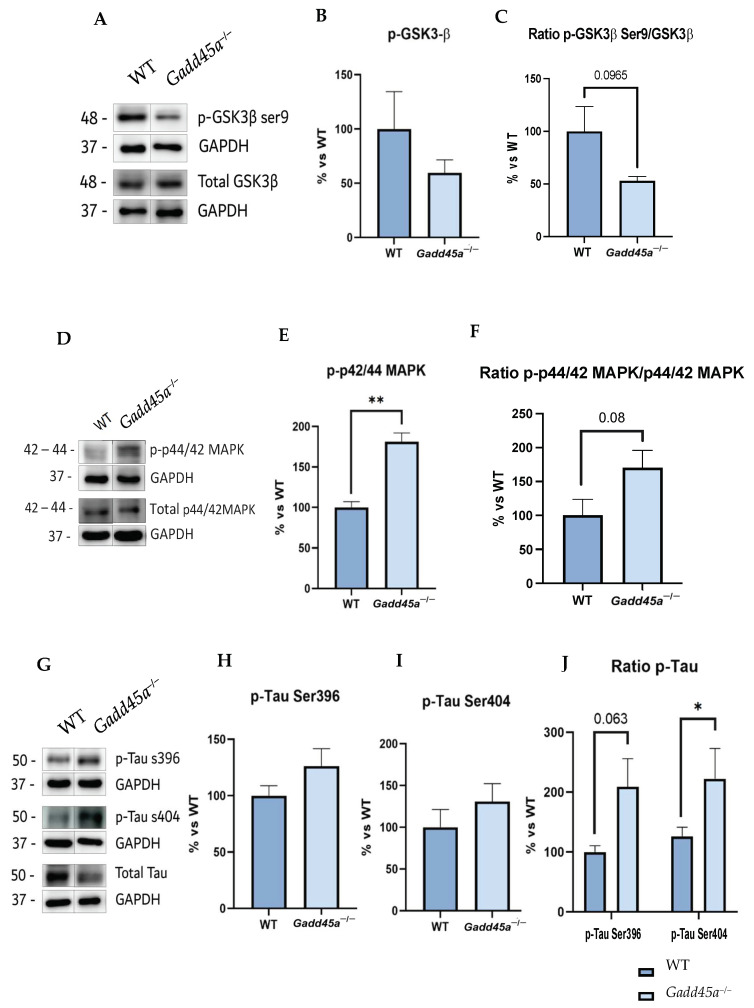
Tau pathology evaluation in *Gadd45a^−/−^* mice. (**A**) Representative WB, and quantification (**B**) of p-GSK3β Ser^9^ and (**C**) ratio of p-GSK3β Ser^9^ and total GSK3β. (**D**) Representative WB, and quantification (**E**) of p-p44/42MAPK and (**F**) ratio p-p44/42MAPK and total p44/42MAPK. (**G**) Representative WB, and quantification (**H**) of p-Tau Ser^396^, (**I**) p-Tau Ser^404^, and (**J**) ratio p-Tau Ser^396^, p-Tau Ser^404^, and total Tau. Bar graph values are 100% adjusted for WT control protein levels. The results are expressed as a mean  ±  SEM. One-tailed Student’s t-test compared groups; (*n* = 4 per group (outliers: *n* = 1 in WT in p-GSK3β ratio levels)); * *p* < 0.05, ** *p* < 0.01.

**Figure 3 ijms-25-02595-f003:**
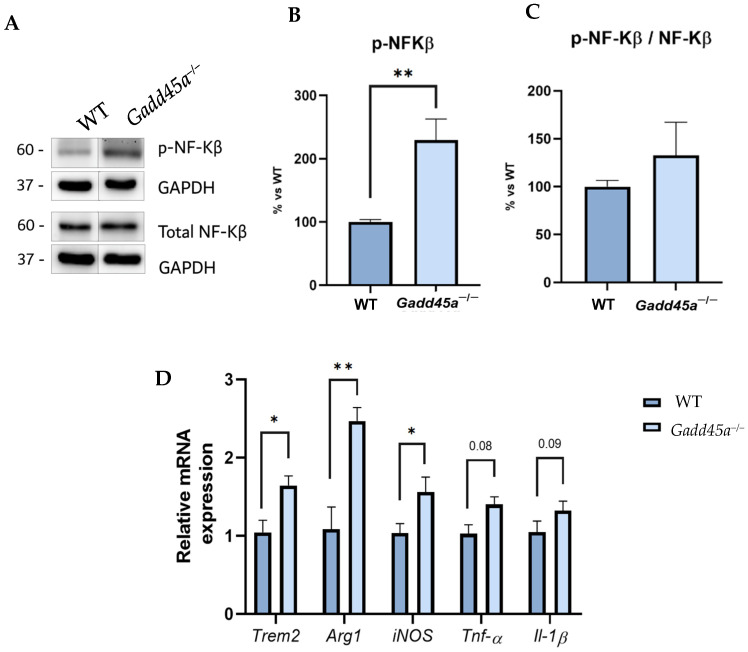
Neuroinflammation markers in *Gadd45a^−/−^* mice. (**A**) Representative WB and quantification (**B**) of p-NF-KB and (**C**) ratio p-NF-KB and total NF-KB. (**D**) Gene expression of inflammatory markers: *Trem2, Arg1, iNOS, Tnf-α,* and *Il-1β.* Gene expression was determined by real-time qPCR. Bar graph values are 100% adjusted for WT control protein level and WT control gene expression, respectively. The results are expressed as a mean ± SEM. One-tailed Student’s *t*-test compared groups; (*n* = 4 per group in (**A**–**C**) and *n* = 6 per group in (**D**) (outliers: *n* = 1 in WT and *n* = 1 in *Gadd45a^−/−^* in *Trem2*; *n* = 1 in WT in *iNOS* and *n* = 2 in WT in *Arg1; n* = 2 in WT and *n* = 2 in *Gadd45a^−/−^* in *Tnf-α;* and *n* = 2 in WT and *n* = 1 in *Il-1β*)); * *p* < 0.05; ** *p* < 0.01.

**Figure 4 ijms-25-02595-f004:**
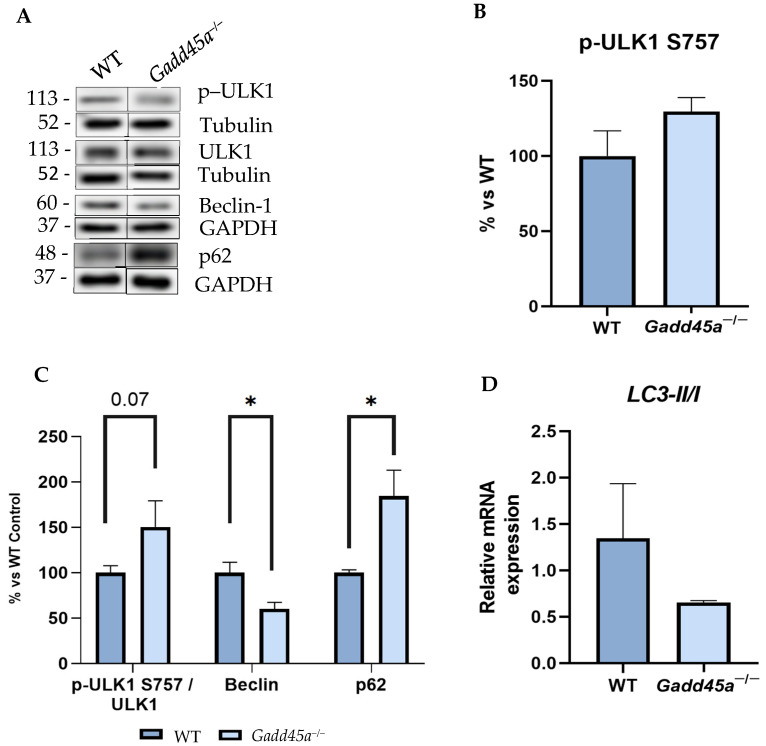
*Gadd45a^−/−^* mice autophagy deregulation. (**A**) Representative WB and quantification (**B**) of p-ULK1 Ser^757^ and (**C**) p-ULK1 Ser^757^/ULK1, Beclin-1, and p62 protein levels. (**D**) Gene expression of LC3II/I was determined by real-time qPCR. Bar graph values are 100% adjusted for WT control protein levels. The results are expressed as a mean  ±  SEM. One-tailed Student’s t-test compared groups (*n* = 4 per group in (**A**–**C**) and *n* = 6 per group in (**D**) (outliers: *n* = 1 in WT in Beclina1; *n* = 1 in WT and *n* = 1 in *Gadd45a^−/-^* in p62; and *n* = 1 in *Gadd45a^−/−^ in LC3-II/I*)); * *p* < 0.05.

**Figure 5 ijms-25-02595-f005:**
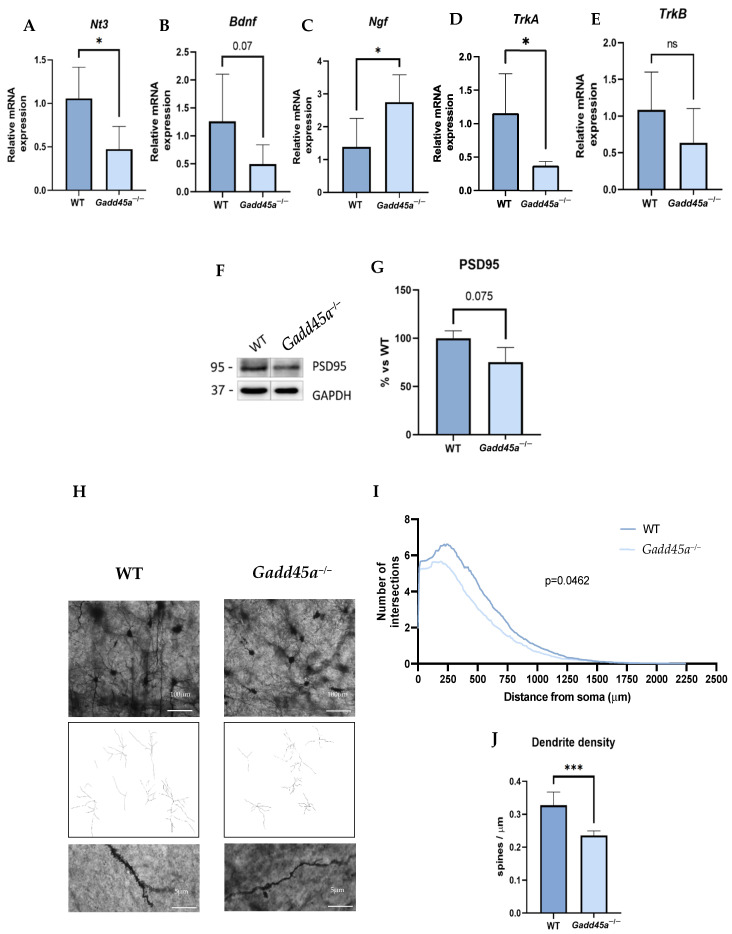
Neural plasticity in *Gadd45a^−/−^* mice: gene expression evaluation of neurotrophins and their receptors: (**A**) *Nt3*, (**B**) *Bdnf,* (**C**) *Ngf,* (**D**) *TrkA,* and (**E**) *TrkB*. (**F**) Representative WB, and quantification (**G**) of PSD95 protein levels. (**H**) Cortex representative images of WT and *Gadd45a^−/−^* Golgi staining (scale bar = 100 µm and 5 µm). (**I**) The number of intersections vs. distance from the soma. (**J**) Quantification of the dendrite density in both groups. Bar graph values are 100% adjusted for WT control gene expression. The results are expressed as a mean  ±  SEM. Groups were compared by one-tailed Student’s t-test; (*n* = 6 per group in (**A**–**E**) (outliers: *n* = 1 in *Gadd45a^−/−^* in *Nt3* expression; *n* = 1 in WT in *TrkA* expression; and *n* = 2 in WT in *Ngf* expression); *n* = 4 per group in (**F**,**G**) and *n* = 100 per group in (**H**–**J**)); * *p* < 0.05; *** *p* < 0.001.

**Figure 6 ijms-25-02595-f006:**
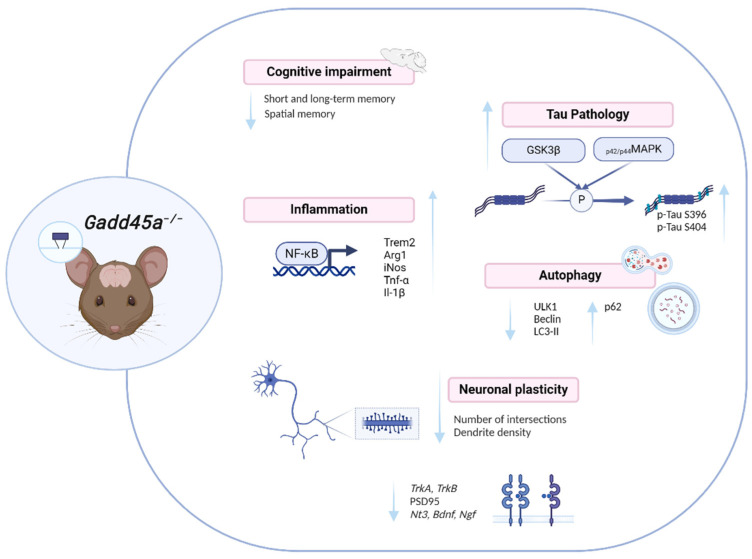
Illustrative scheme of the *Gadd45a^−/−^* effects.

**Figure 7 ijms-25-02595-f007:**
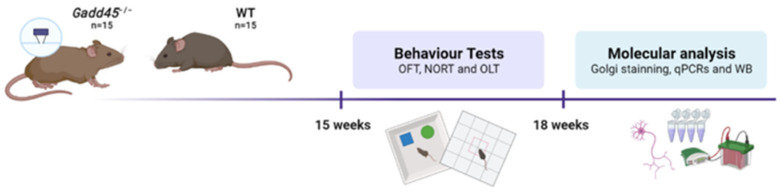
Experimental design scheme.

## Data Availability

Data are contained within the article and Appendix A.

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
