# Peer review of "Deletion of Gadd45a Expression in Mice Leads to Cognitive and Synaptic Impairment Associated with Alzheimer’s Disease Hallmarks"

_ijms, 2024, doi:10.3390/ijms25052595_

Round 1
Reviewer 1 Report
Comments and Suggestions for Authors
This manuscript presented some interesting findings and phenotypes associated with Alzheimer’s disease in the brain of mice that lack of Gadd45a gene. Several concerns are raised regarding the data illustrated in this study:
1. Their data showed an increase in Tau hyperphosphorylation and in the levels of kinases involved in its phosphorylation in the hippocampus. Moreover, Gadd45a-/- animals showed a significant increase in pro-inflammatory cytokines as well as autophagy markers in the brain.
In this study, the pathological hallmarks related to AD were observed by analyzing whole brain tissue. However, these hallmarks are mixed observation; the authors did not distinguish between cell type-specific alterations (neurons, astrocytes and microglia). Some alterations may come from interactions between different cell types. So, it is hard to interpret and understand the exact role of Gadd45a gene, particularly in terms of signaling pathways.
2. Line 133-151: The authors evaluated those proinflammatory target genes such as triggering receptor expressed on myeloid cells 2 (Trem2), arginase 1 (Arg1) and inducible nitric 140 oxide synthase (iNOS). However, they didn’t show any changes in the pro-inflammatory cytokines, such as TNF-α、IL-1β and IL-6. This assay is very important to illustrate inflammatory phenotypes. In addition, the levels of NF-κB and the genes of Trem2, Arg1 and iNOS were upregulate. The question is what types of cells (neurons, astrocytes or microglia?) contributed to this upregulation.
3. Line 152-157: The data show a reduction in Autophagy markers. This raised the same question as what types of cells (neurons, astrocytes or microglia?) contributed to this reduction.
4. Line 190-191: What parts of the brain was analyzed by the Golgi Staining, Cortex or hippocampus?
Comments on the Quality of English LanguageEnglish language is fine.
Author Response
Reviewer 1
1.This manuscript presented some interesting findings and phenotypes associated with Alzheimer’s disease in the brain of mice that lack of Gadd45a gene. Several concerns are raised regarding the data illustrated in this study:
Their data showed an increase in Tau hyperphosphorylation and in the levels of kinases involved in its phosphorylation in the hippocampus. Moreover, Gadd45a-/- animals showed a significant increase in pro-inflammatory cytokines as well as autophagy markers in the brain.
In this study, the pathological hallmarks related to AD were observed by analyzing whole brain tissue. However, these hallmarks are mixed observation; the authors did not distinguish between cell type-specific alterations (neurons, astrocytes and microglia). Some alterations may come from interactions between different cell types. So, it is hard to interpret and understand the exact role of Gadd45a gene, particularly in terms of signaling pathways.
Answer: Thanks to the reviewer for this observation. Unfortunately, we cannot distinguish between cell type-specific alterations because we have processed the whole brain to perform different techniques such as western blotting, qPCR, and Golgi staining. Thereby, the main issue to answer was described as the main features after the deletion of GADD45A. This was not the objective of the present manuscript. However, as we observed those changes, our next propose will be to evaluate these differences at the cell type level.
- Line 133-151: The authors evaluated those proinflammatory target genes such as triggering receptor expressed on myeloid cells 2 (Trem2), arginase 1 (Arg1) and inducible nitric 140 oxide synthase (iNOS). However, they didn’t show any changes in the pro-inflammatory cytokines, such as TNF-α、IL-1β and IL-6. This assay is very important to illustrate inflammatory phenotypes. In addition, the levels of NF-κB and the genes of Trem2, Arg1 and iNOS were upregulate. The question is what types of cells (neurons, astrocytes or microglia?) contributed to this upregulation.
Answer: Thanks to the reviewer for this observation. Accordingly, we have added Tnf-a and Il-1β gene levels, and we obtained differences between WT and GADD45a-/- mice, confirming our previous hypothesis. Unfortunately, we cannot distinguish between cell type-specific alterations because we have processed and analyzed the whole brain.
- Line 152-157: The data show a reduction in Autophagy markers. This raised the same question as what types of cells (neurons, astrocytes or microglia?) contributed to this reduction.
Answer: Thanks to the reviewer for this observation. Unfortunately, we cannot distinguish between cell type-specific alterations because we have processed and analyzed the whole brain. However, we have included some more data in order to improve the quality of the results section and our hypothesis regarding the autophagy alterations presented in this manuscript.
- Line 190-191: What parts of the brain were analyzed by the Golgi Staining, Cortex, or hippocampus?
Answer: Thanks to the reviewer for this observation. We have changed the text accordingly: “Considering these results, we decided to analyze the complexity of the neurons and the dendrite density located on the cortex of WT and KO brain animals by Golgi Staining.”
Reviewer 2
The manuscript by Christian Grinan Ferre et al, titled "Deletion of Gadd45a Expression in Mice Leads to Cognitive and Synaptic Impairment Associated with Alzheimer’s Disease Hallmarks," highlights that the absence of the Gadd45a gene triggers various pathways that worsen Alzheimer's Disease (AD) pathology. This suggests that enhancing the expression or function of this protein could be a promising therapeutic approach to decelerate the progression of AD.
While the paper is intriguing, there are certain points that need to be addressed before publication.
Introduction: the first part need to be rewritten in a better way: Es: The aging process unfolds as a gradual pathophysiological journey, precipitating a diminishing state of physical and cognitive functions throughout all organs. This progressive decline is underscored by the accrual of damage in response to diverse insults and stressors.
Answer: Thanks to the reviewer for this observation. Accordingly, we have added it in the introduction.
Line 52: In this context, the authors could enrich their introduction by incorporating additional references related to autophagy and its relevance to the Alzheimer's Disease (AD) brain.
Answer: Thanks to the reviewer for this observation. We have added this in the introduction and some more in the discussion: “Specifically, autophagy is a relevant intracellular self-degradative process that degrades and recycles cellular components such as dysfunctional organelles and abnormally aggregated and misfolded proteins [11]. Interestingly, Tau pathology depends on the autophagy process and this mechanism is dysregulated in AD patients, facilitating protein aggregation and disease progression [12].”
Line 70: It's essential for the authors to supplement their assertions in this section with relevant references to strengthen the credibility and comprehensiveness of their arguments.
Answer: Thanks to the reviewer for this observation. Accordingly, we have added some relevant references.
Results Section: The incorporation of high-resolution images is warranted to enhance the clarity and precision of the findings.
Answer: Thanks the reviewer for noticing this. We have revised the images and we improved the resolution.
Image Quality: The images provided do not meet the standards for optimal resolution, resulting in reduced clarity. Specifically, in Figure 1, the letter 'E' is not distinctly visible and needs improvement to ensure accurate interpretation. For the publication the authors over that the ratio of the protein should insert also the phosphorylation alone.
Answer: Thanks the reviewer for noticing this. We have corrected the letter E in Figure 1 accordingly and added the protein phosphorylation graph alone.
To comprehensively characterize autophagy, the authors are encouraged to include additional experiments in the paper.
Answer: Thanks the reviewer for this observation. Accordingly, we have added p62 protein levels and LC3II/I gene expression and we have improved the results and discussion section. Now, the results confirm the idea that the autophagy process is altered in this mice model and partially explain the neurodegenerative process presented in this mouse.
Immunofluorescence Microscopy: Utilize immunofluorescence staining to visualize the subcellular localization of autophagy-related proteins within cells. This can offer spatial information and confirm the presence of autophagic structures.
Answer: Thanks the reviewer for this observation. It was impossible to do IHQ, because we performed Western blotting, qPCR with these two techniques we used half hemisphere and the other was for Golgi staining. However, to accomplish with your suggestions, we have added a western blot of p62 with an important changes in GADD45a-/- mice model and LC3II/I as well.
mTOR Activity Assays: Investigate mTOR signaling pathway activity, as it plays a crucial role in regulating autophagy. This can be done through the measurement of mTOR phosphorylation or downstream targets.
Autophagy Gene Expression Analysis: Analyze the expression levels of key autophagy-related genes using quantitative PCR (qPCR). This can provide additional information about the regulation of autophagy. Including a combination of these experiments can provide a more comprehensive understanding of the autophagic process, complementing the Western blot analysis of ULK1 and Beclin-1.
Answer: Thanks to the reviewer for this observation. Accordingly, we have added a western blot of p62 with important changes in the GADD45a-/- mice model and LC3II/I as well.
In the supplementary material, the authors are advised to include all uncropped figures, including Figure 3 and Figure 4, which were previously omitted. Kindly submit a revised file containing all figures for completeness.
Answer: Thanks to the reviewer for this observation. We have included all the images.
The manuscript cannot be accepted in its current form; it requires thorough and precise revision for consideration.
Comments on the Quality of English Language
The quality of the English language in the manuscript is of median quality. While the overall language is understandable, there is room for improvement in terms of clarity, precision, and adherence to academic writing conventions. Addressing specific grammatical issues, refining sentence structures, and ensuring consistent terminology could enhance the overall linguistic quality of the manuscript.
Answer: Thanks to the reviewer for this observation. Accordingly, we have improved the linguistic quality of the manuscript, as you can see in the new version.

Reviewer 2 Report
Comments and Suggestions for Authors
The manuscript by Christian Grinan Ferre et al, titled "Deletion of Gadd45a Expression in Mice Leads to Cognitive and Synaptic Impairment Associated with Alzheimer’s Disease Hallmarks," highlights that the absence of the Gadd45a gene triggers various pathways that worsen Alzheimer's Disease (AD) pathology. This suggests that enhancing the expression or function of this protein could be a promising therapeutic approach to decelerate the progression of AD.
While the paper is intriguing, there are certain points that need to be addressed before publication.
Introduction: the first part need to be rewritten in a better way: Es: The aging process unfolds as a gradual pathophysiological journey, precipitating a diminishing state of physical and cognitive functions throughout all organs. This progressive decline is underscored by the accrual of damage in response to diverse insults and stressors.
Line 52: In this context, the authors could enrich their introduction by incorporating additional references related to autophagy and its relevance to the Alzheimer's Disease (AD) brain.
Line 70: It's essential for the authors to supplement their assertions in this section with relevant references to strengthen the credibility and comprehensiveness of their arguments.
Results Section: The incorporation of high-resolution images is warranted to enhance the clarity and precision of the findings.
Image Quality: The images provided do not meet the standards for optimal resolution, resulting in reduced clarity. Specifically, in Figure 1, the letter 'E' is not distinctly visible and needs improvement to ensure accurate interpretation.
For the publication the authors over that the ratio of the protein should insert also the phosphorylation alone.
To comprehensively characterize autophagy, the authors are encouraged to include additional experiments in the paper.
Immunofluorescence Microscopy: Utilize immunofluorescence staining to visualize the subcellular localization of autophagy-related proteins within cells. This can offer spatial information and confirm the presence of autophagic structures.
mTOR Activity Assays: Investigate mTOR signaling pathway activity, as it plays a crucial role in regulating autophagy. This can be done through the measurement of mTOR phosphorylation or downstream targets.
Autophagy Gene Expression Analysis: Analyze the expression levels of key autophagy-related genes using quantitative PCR (qPCR). This can provide additional information about the regulation of autophagy.
Including a combination of these experiments can provide a more comprehensive understanding of the autophagic process, complementing the Western blot analysis of ULK1 and Beclin-1.
In the supplementary material, the authors are advised to include all uncropped figures, including Figure 3 and Figure 4, which were previously omitted. Kindly submit a revised file containing all figures for completeness.
The manuscript cannot be accepted in its current form; it requires thorough and precise revision for consideration.
Comments on the Quality of English LanguageThe quality of the English language in the manuscript is of median quality. While the overall language is understandable, there is room for improvement in terms of clarity, precision, and adherence to academic writing conventions. Addressing specific grammatical issues, refining sentence structures, and ensuring consistent terminology could enhance the overall linguistic quality of the manuscript.
Author Response
Reviewer 1
1.This manuscript presented some interesting findings and phenotypes associated with Alzheimer’s disease in the brain of mice that lack of Gadd45a gene. Several concerns are raised regarding the data illustrated in this study:
Their data showed an increase in Tau hyperphosphorylation and in the levels of kinases involved in its phosphorylation in the hippocampus. Moreover, Gadd45a-/- animals showed a significant increase in pro-inflammatory cytokines as well as autophagy markers in the brain.
In this study, the pathological hallmarks related to AD were observed by analyzing whole brain tissue. However, these hallmarks are mixed observation; the authors did not distinguish between cell type-specific alterations (neurons, astrocytes and microglia). Some alterations may come from interactions between different cell types. So, it is hard to interpret and understand the exact role of Gadd45a gene, particularly in terms of signaling pathways.
Answer: Thanks to the reviewer for this observation. Unfortunately, we cannot distinguish between cell type-specific alterations because we have processed the whole brain to perform different techniques such as western blotting, qPCR, and Golgi staining. Thereby, the main issue to answer was described as the main features after the deletion of GADD45A. This was not the objective of the present manuscript. However, as we observed those changes, our next propose will be to evaluate these differences at the cell type level.
- Line 133-151: The authors evaluated those proinflammatory target genes such as triggering receptor expressed on myeloid cells 2 (Trem2), arginase 1 (Arg1) and inducible nitric 140 oxide synthase (iNOS). However, they didn’t show any changes in the pro-inflammatory cytokines, such as TNF-α、IL-1β and IL-6. This assay is very important to illustrate inflammatory phenotypes. In addition, the levels of NF-κB and the genes of Trem2, Arg1 and iNOS were upregulate. The question is what types of cells (neurons, astrocytes or microglia?) contributed to this upregulation.
Answer: Thanks to the reviewer for this observation. Accordingly, we have added Tnf-a and Il-1β gene levels, and we obtained differences between WT and GADD45a-/- mice, confirming our previous hypothesis. Unfortunately, we cannot distinguish between cell type-specific alterations because we have processed and analyzed the whole brain.
- Line 152-157: The data show a reduction in Autophagy markers. This raised the same question as what types of cells (neurons, astrocytes or microglia?) contributed to this reduction.
Answer: Thanks to the reviewer for this observation. Unfortunately, we cannot distinguish between cell type-specific alterations because we have processed and analyzed the whole brain. However, we have included some more data in order to improve the quality of the results section and our hypothesis regarding the autophagy alterations presented in this manuscript.
- Line 190-191: What parts of the brain were analyzed by the Golgi Staining, Cortex, or hippocampus?
Answer: Thanks to the reviewer for this observation. We have changed the text accordingly: “Considering these results, we decided to analyze the complexity of the neurons and the dendrite density located on the cortex of WT and KO brain animals by Golgi Staining.”
Reviewer 2
The manuscript by Christian Grinan Ferre et al, titled "Deletion of Gadd45a Expression in Mice Leads to Cognitive and Synaptic Impairment Associated with Alzheimer’s Disease Hallmarks," highlights that the absence of the Gadd45a gene triggers various pathways that worsen Alzheimer's Disease (AD) pathology. This suggests that enhancing the expression or function of this protein could be a promising therapeutic approach to decelerate the progression of AD.
While the paper is intriguing, there are certain points that need to be addressed before publication.
Introduction: the first part need to be rewritten in a better way: Es: The aging process unfolds as a gradual pathophysiological journey, precipitating a diminishing state of physical and cognitive functions throughout all organs. This progressive decline is underscored by the accrual of damage in response to diverse insults and stressors.
Answer: Thanks to the reviewer for this observation. Accordingly, we have added it in the introduction.
Line 52: In this context, the authors could enrich their introduction by incorporating additional references related to autophagy and its relevance to the Alzheimer's Disease (AD) brain.
Answer: Thanks to the reviewer for this observation. We have added this in the introduction and some more in the discussion: “Specifically, autophagy is a relevant intracellular self-degradative process that degrades and recycles cellular components such as dysfunctional organelles and abnormally aggregated and misfolded proteins [11]. Interestingly, Tau pathology depends on the autophagy process and this mechanism is dysregulated in AD patients, facilitating protein aggregation and disease progression [12].”
Line 70: It's essential for the authors to supplement their assertions in this section with relevant references to strengthen the credibility and comprehensiveness of their arguments.
Answer: Thanks to the reviewer for this observation. Accordingly, we have added some relevant references.
Results Section: The incorporation of high-resolution images is warranted to enhance the clarity and precision of the findings.
Answer: Thanks the reviewer for noticing this. We have revised the images and we improved the resolution.
Image Quality: The images provided do not meet the standards for optimal resolution, resulting in reduced clarity. Specifically, in Figure 1, the letter 'E' is not distinctly visible and needs improvement to ensure accurate interpretation. For the publication the authors over that the ratio of the protein should insert also the phosphorylation alone.
Answer: Thanks the reviewer for noticing this. We have corrected the letter E in Figure 1 accordingly and added the protein phosphorylation graph alone.
To comprehensively characterize autophagy, the authors are encouraged to include additional experiments in the paper.
Answer: Thanks the reviewer for this observation. Accordingly, we have added p62 protein levels and LC3II/I gene expression and we have improved the results and discussion section. Now, the results confirm the idea that the autophagy process is altered in this mice model and partially explain the neurodegenerative process presented in this mouse.
Immunofluorescence Microscopy: Utilize immunofluorescence staining to visualize the subcellular localization of autophagy-related proteins within cells. This can offer spatial information and confirm the presence of autophagic structures.
Answer: Thanks the reviewer for this observation. It was impossible to do IHQ, because we performed Western blotting, qPCR with these two techniques we used half hemisphere and the other was for Golgi staining. However, to accomplish with your suggestions, we have added a western blot of p62 with an important changes in GADD45a-/- mice model and LC3II/I as well.
mTOR Activity Assays: Investigate mTOR signaling pathway activity, as it plays a crucial role in regulating autophagy. This can be done through the measurement of mTOR phosphorylation or downstream targets.
Autophagy Gene Expression Analysis: Analyze the expression levels of key autophagy-related genes using quantitative PCR (qPCR). This can provide additional information about the regulation of autophagy. Including a combination of these experiments can provide a more comprehensive understanding of the autophagic process, complementing the Western blot analysis of ULK1 and Beclin-1.
Answer: Thanks to the reviewer for this observation. Accordingly, we have added a western blot of p62 with important changes in the GADD45a-/- mice model and LC3II/I as well.
In the supplementary material, the authors are advised to include all uncropped figures, including Figure 3 and Figure 4, which were previously omitted. Kindly submit a revised file containing all figures for completeness.
Answer: Thanks to the reviewer for this observation. We have included all the images.
The manuscript cannot be accepted in its current form; it requires thorough and precise revision for consideration.
Comments on the Quality of English Language
The quality of the English language in the manuscript is of median quality. While the overall language is understandable, there is room for improvement in terms of clarity, precision, and adherence to academic writing conventions. Addressing specific grammatical issues, refining sentence structures, and ensuring consistent terminology could enhance the overall linguistic quality of the
Reviewer 1
1.This manuscript presented some interesting findings and phenotypes associated with Alzheimer’s disease in the brain of mice that lack of Gadd45a gene. Several concerns are raised regarding the data illustrated in this study:
Their data showed an increase in Tau hyperphosphorylation and in the levels of kinases involved in its phosphorylation in the hippocampus. Moreover, Gadd45a-/- animals showed a significant increase in pro-inflammatory cytokines as well as autophagy markers in the brain.
In this study, the pathological hallmarks related to AD were observed by analyzing whole brain tissue. However, these hallmarks are mixed observation; the authors did not distinguish between cell type-specific alterations (neurons, astrocytes and microglia). Some alterations may come from interactions between different cell types. So, it is hard to interpret and understand the exact role of Gadd45a gene, particularly in terms of signaling pathways.
Answer: Thanks to the reviewer for this observation. Unfortunately, we cannot distinguish between cell type-specific alterations because we have processed the whole brain to perform different techniques such as western blotting, qPCR, and Golgi staining. Thereby, the main issue to answer was described as the main features after the deletion of GADD45A. This was not the objective of the present manuscript. However, as we observed those changes, our next propose will be to evaluate these differences at the cell type level.
- Line 133-151: The authors evaluated those proinflammatory target genes such as triggering receptor expressed on myeloid cells 2 (Trem2), arginase 1 (Arg1) and inducible nitric 140 oxide synthase (iNOS). However, they didn’t show any changes in the pro-inflammatory cytokines, such as TNF-α、IL-1β and IL-6. This assay is very important to illustrate inflammatory phenotypes. In addition, the levels of NF-κB and the genes of Trem2, Arg1 and iNOS were upregulate. The question is what types of cells (neurons, astrocytes or microglia?) contributed to this upregulation.
Answer: Thanks to the reviewer for this observation. Accordingly, we have added Tnf-a and Il-1β gene levels, and we obtained differences between WT and GADD45a-/- mice, confirming our previous hypothesis. Unfortunately, we cannot distinguish between cell type-specific alterations because we have processed and analyzed the whole brain.
- Line 152-157: The data show a reduction in Autophagy markers. This raised the same question as what types of cells (neurons, astrocytes or microglia?) contributed to this reduction.
Answer: Thanks to the reviewer for this observation. Unfortunately, we cannot distinguish between cell type-specific alterations because we have processed and analyzed the whole brain. However, we have included some more data in order to improve the quality of the results section and our hypothesis regarding the autophagy alterations presented in this manuscript.
- Line 190-191: What parts of the brain were analyzed by the Golgi Staining, Cortex, or hippocampus?
Answer: Thanks to the reviewer for this observation. We have changed the text accordingly: “Considering these results, we decided to analyze the complexity of the neurons and the dendrite density located on the cortex of WT and KO brain animals by Golgi Staining.”
Reviewer 2
The manuscript by Christian Grinan Ferre et al, titled "Deletion of Gadd45a Expression in Mice Leads to Cognitive and Synaptic Impairment Associated with Alzheimer’s Disease Hallmarks," highlights that the absence of the Gadd45a gene triggers various pathways that worsen Alzheimer's Disease (AD) pathology. This suggests that enhancing the expression or function of this protein could be a promising therapeutic approach to decelerate the progression of AD.
While the paper is intriguing, there are certain points that need to be addressed before publication.
Introduction: the first part need to be rewritten in a better way: Es: The aging process unfolds as a gradual pathophysiological journey, precipitating a diminishing state of physical and cognitive functions throughout all organs. This progressive decline is underscored by the accrual of damage in response to diverse insults and stressors.
Answer: Thanks to the reviewer for this observation. Accordingly, we have added it in the introduction.
Line 52: In this context, the authors could enrich their introduction by incorporating additional references related to autophagy and its relevance to the Alzheimer's Disease (AD) brain.
Answer: Thanks to the reviewer for this observation. We have added this in the introduction and some more in the discussion: “Specifically, autophagy is a relevant intracellular self-degradative process that degrades and recycles cellular components such as dysfunctional organelles and abnormally aggregated and misfolded proteins [11]. Interestingly, Tau pathology depends on the autophagy process and this mechanism is dysregulated in AD patients, facilitating protein aggregation and disease progression [12].”
Line 70: It's essential for the authors to supplement their assertions in this section with relevant references to strengthen the credibility and comprehensiveness of their arguments.
Answer: Thanks to the reviewer for this observation. Accordingly, we have added some relevant references.
Results Section: The incorporation of high-resolution images is warranted to enhance the clarity and precision of the findings.
Answer: Thanks the reviewer for noticing this. We have revised the images and we improved the resolution.
Image Quality: The images provided do not meet the standards for optimal resolution, resulting in reduced clarity. Specifically, in Figure 1, the letter 'E' is not distinctly visible and needs improvement to ensure accurate interpretation. For the publication the authors over that the ratio of the protein should insert also the phosphorylation alone.
Answer: Thanks the reviewer for noticing this. We have corrected the letter E in Figure 1 accordingly and added the protein phosphorylation graph alone.
To comprehensively characterize autophagy, the authors are encouraged to include additional experiments in the paper.
Answer: Thanks the reviewer for this observation. Accordingly, we have added p62 protein levels and LC3II/I gene expression and we have improved the results and discussion section. Now, the results confirm the idea that the autophagy process is altered in this mice model and partially explain the neurodegenerative process presented in this mouse.
Immunofluorescence Microscopy: Utilize immunofluorescence staining to visualize the subcellular localization of autophagy-related proteins within cells. This can offer spatial information and confirm the presence of autophagic structures.
Answer: Thanks the reviewer for this observation. It was impossible to do IHQ, because we performed Western blotting, qPCR with these two techniques we used half hemisphere and the other was for Golgi staining. However, to accomplish with your suggestions, we have added a western blot of p62 with an important changes in GADD45a-/- mice model and LC3II/I as well.
mTOR Activity Assays: Investigate mTOR signaling pathway activity, as it plays a crucial role in regulating autophagy. This can be done through the measurement of mTOR phosphorylation or downstream targets.
Autophagy Gene Expression Analysis: Analyze the expression levels of key autophagy-related genes using quantitative PCR (qPCR). This can provide additional information about the regulation of autophagy. Including a combination of these experiments can provide a more comprehensive understanding of the autophagic process, complementing the Western blot analysis of ULK1 and Beclin-1.
Answer: Thanks to the reviewer for this observation. Accordingly, we have added a western blot of p62 with important changes in the GADD45a-/- mice model and LC3II/I as well.
In the supplementary material, the authors are advised to include all uncropped figures, including Figure 3 and Figure 4, which were previously omitted. Kindly submit a revised file containing all figures for completeness.
Answer: Thanks to the reviewer for this observation. We have included all the images.
The manuscript cannot be accepted in its current form; it requires thorough and precise revision for consideration.
Comments on the Quality of English Language
The quality of the English language in the manuscript is of median quality. While the overall language is understandable, there is room for improvement in terms of clarity, precision, and adherence to academic writing conventions. Addressing specific grammatical issues, refining sentence structures, and ensuring consistent terminology could enhance the overall linguistic quality of the manuscript.
Answer: Thanks to the reviewer for this observation. Accordingly, we have improved the linguistic quality of the manuscript, as you can see in the new version.
.
Answer: Thanks to the reviewer for this observation. Accordingly, we have improved the linguistic quality of the manuscript, as you can see in the new version.

Round 2
Reviewer 1 Report
Comments and Suggestions for Authors
The revision is fine.